# Preparation of Low-Molecular-Weight Fucoidan with Anticoagulant Activity by Photocatalytic Degradation Method

**DOI:** 10.3390/foods11060822

**Published:** 2022-03-13

**Authors:** Yihui Qi, Lilong Wang, Ying You, Xiaona Sun, Chengrong Wen, Yinghuan Fu, Shuang Song

**Affiliations:** 1National Engineering Research Center of Seafood, School of Food Science and Technology, Dalian Polytechnic University, Dalian 116034, China; qiyh0420@163.com (Y.Q.); wanglilonghn@163.com (L.W.); 18946657507@163.com (Y.Y.); xiaonana1023@126.com (X.S.); wencr2014@163.com (C.W.); fuyh@dlpu.edu.cn (Y.F.); 2National & Local Joint Engineering Laboratory for Marine Bioactive Polysaccharide Development and Application, Dalian Polytechnic University, Dalian 116034, China; 3College of Food Science and Engineering, Jilin Agricultural University, Changchun 130117, China; 4School of Light Industry and Chemical Engineering, Dalian Polytechnic University, Dalian 116034, China

**Keywords:** fucoidan, photocatalytic degradation, anticoagulant activity

## Abstract

It is a challenge to degrade sulfated polysaccharides without stripping sulfate groups. In the present study, a photocatalytic method was applied to degrade fucoidan, a sulfated polysaccharide from brown algae. The degradation with varying addition amounts of H_2_O_2_ and TiO_2_ were monitored by high performance gel permeation chromatography (HPGPC) and thin layer chromatography (TLC), and fucoidan was efficiently degraded with 5% TiO_2_ and 0.95% H_2_O_2_. A comparison of the chemical compositions of 2 products obtained after 0.5 h and 3 h illumination, DF-0.5 (average Mw 90 kDa) and DF-3 (average Mw 3 kDa), respectively, with those of fucoidan indicates the photocatalytic degradation did not strip the sulfate groups, but reduced the galactose/fucose ratio. Moreover, 12 oligosaccharides in DF-3 were identified by HPLC-ESI-MS^n^ and 10 of them were sulfated. In addition, DF-0.5 showed anticoagulant activity as strong as fucoidan while DF-3 could specifically prolong the activated partial thromboplastin time. All samples exerted inhibition effects on the intrinsic pathway FXII in a dose-dependent manner. Thus, photocatalytic degradation demonstrated the potential to prepare sulfated low-molecular-weight fucoidan with anticoagulant activity.

## 1. Introduction

Fucoidans are water-soluble biologically active sulfated polysaccharides in brown algae [1], which are mainly composed of L-fucose and small amounts of galactose, glucose, xylose, mannose, and uronic acid. There are numerous reports of fucoidans that have a number of biological activities, including anti-coagulant [2,3], anti-tumor [4], anti-inflammatory [5], anti-diabetic [6], anti-oxidant [7], anti-viral [8], and anti-thrombotic [9] effects. The molecular weight of fucoidan is related to its biological activity and the low molecular weight fucoidan usually has better biological activity [2,9]. The researchers found that the inhibitory effects of fucoidans on coagulation and cell proliferation are dependent on their sulfation degree [9]. Other researchers also found that the low molecular weight of fucoidan has higher anticoagulant activity in vivo, and the sulfate content plays an important role on anticoagulant activity [10]. Other researchers have also found that low molecular weight fucoidan is easier to be degraded and utilized by intestinal microorganisms [11]. Therefore, it is indeed to find an efficient and safe method for fucoidan degradation.

Traditional degradation methods for fucoidans include acid hydrolysis [12], enzymatic degradation [13], ultrasonic assisted degradation [14], and microwave-assisted degradation [15], etc. Acid hydrolysis has often been applied to degrade polysaccharides, but it could cause environmental pollution. Moreover, it could damage the sulfate group of fucoidan leading to the loss of bioactivity [16]. The enzymatic approach for fucoidan degradation is high-cost and inefficient which has restricted its industrial-scale applications [17]. Ultrasonic assisted degradation and microwave-assisted degradation have high requirements for equipment, which greatly increases the production cost of fucoidan degradation [18,19]. Therefore, it is necessary to find an environmentally friendly, low-cost, high-efficiency degradation method which can retain the sulfate group of fucoidan.

Photocatalytic degradation, as an emerging ‘Green Chemistry’ technology, has attracted increasing attention in recent years. Photochemical oxidation processes have been used for degradation of many different organic pollutants [20]. This process is simpler, higher yielding and more environmentally benign than conventional methods. The photocatalytic process originates from the semiconductor band gap. The photons with a higher energy than the band gap can be absorbed while an electron is promoted to the conduction band, leaving a hole in the valence band. Then this excited electron can be used to drive a degradation reaction [21]. This degradation performance is attributed to highly oxidizing holes and hydroxyl radicals (HO•) that are known as strong oxidizers [21]. TiO_2_ is the most widely used catalyst for photocatalytic degradation because of its chemical stability and low biological toxicity [22]. Therefore, photocatalytic degradation shows advantages of low-energy-cost and high-degradation-efficiency. However, there is no report on its application in the polymerization of sulfated polysaccharides.

Thus, the present study investigated the TiO_2_-catalyzed photocatalytic reaction for fucoidan degradation, and the degradation products were characterized and compared with the undegraded fucoidan by a series of analysis techniques to reveal the compositional, structural, cytotoxicity and anticoagulant activity changes of polysaccharides caused by photocatalytic degradation.

## 2. Materials and Methods

### 2.1. Materials

Fucoidan from *Undaria pinnatifida* was supplied by Qingdao Bright Moon Seaweed Group Co., Ltd. (Qingdao, China). Its average molecular weight was determined as 190 kDa and its sulfate content as 26%. Titanium dioxide (TiO_2_) nanoparticles (25 nm) were purchased from Evonik Degussa China Co., Ltd. (Shanghai, China) (Appendix A) (Appendix A). Acetonitrile and methanol alcohol as chromatographic grade were purchased from Fisher, Pittsburgh, PA, USA.

### 2.2. Photocatalytic Degradation of Fucoidan

Photocatalytic Xenon Light Source System (CEL-HXF300-T3) was purchased from Beijing Zhongjiao Jinyuan Technology Co., Ltd. (Beijing, China) Photocatalytic reaction apparatus consists of a light source (Xenon lamp 300 W, PerkinElmer), a filter (AREF 300–1100 nm) and a power source.

The photocatalytic reaction was carried out in a reactor with dimension of 70 × 50 mm (height × diameter). The irradiation was carried out using a 300 W Xenon lamp. Fucoidan (1.0 g) was dissolved in water (200 mL) mixture at room temperature under magnetic stirring for 12 h to obtain a homogeneous fucoidan solution. Titanium dioxide (0.2–1.0 g) was added into the fucoidan solution. A magnetic stirrer was located at the reactor’s base, which a homogenous TiO_2_ suspension could be maintained throughout the reaction. Immediately after the addition of H_2_O_2_ (0–6.4 mL) to the reactor, the Xenon light was turned on to initiate the reaction (pH 6). During the photolysis experiments, the solution composed of the fucoidan solution and catalyst was placed in the reactor and stirred magnetically with simultaneous exposure to Xenon light and the illumination time was between 0.5 h and 9 h. After the reaction, the solution was centrifuged at 9600× *g* for 10 min and the supernatant solution was dialyzed (molecular weight cut-off 300 Da) and lyophilized (chamber pressure 15 Pa, cold trap temperature −45 °C and material thickness 10 mm).

### 2.3. Chemical Analysis

The total sugar content of the degraded polysaccharide was determined by the phenol-sulfuric acid method at 490 nm using fucose as standard [23]. The sulfate group content was determined by the BaCl_2_-gelatin turbidity method at 360 nm using K_2_SO_4_ as standard [24]. The content of sulfated polysaccharide was determined by the metachromatic assay with 1,9-dimethylmethylene blue (DMB) [25] at 525 nm by UV-visible spectrophotometer, using fucoidan as the standard. The calculation defined for the contents of sulfate group and sulfated polysaccharide are shown in Appendix A.

### 2.4. Determination of Molecular Weight Distribution

The average molecular weight of degraded polysaccharide was determined by high performance gel permeation chromatography (HPGPC) using chromatographic column of TSK-4000PWXL (7.8 mm × 300 mm) and TSK-5000PWXL (7.8 mm × 300 mm). The effluent was monitored by a Waters 2414 refractive index detector. The sample was centrifuged at 9600× *g* for 10 min and filtrated on 0.22 μm filter membrane. The injection volume was 10 μL, and the mobile phase was 0.1 M ammonium acetate at 0.4 mL/min with the column temperature at 30 °C. Different molecular weight dextrans (1, 5, 12, 25, 50, 150, 410, and 670 kDa) were used as molecular weight standards. The dextrans were purchased from Sigma Chemical Co., Ltd. (Shanghai, China).

### 2.5. Thin Layer Chromatography (TLC)

The degraded polysaccharide sample (0, 0.5, 1, 1.5, 2, 2.5, 3, 4, 5 and 6 h) and standards (glucose, lactose, β-cyclodextrin) were injected onto the TLC plate (Merck, Germany). The separation was performed with n-butanol, glacial acetic acid, and water (2:1:1, *v*/*v*/*v*). Then the TLC plate was dried for 15 min at room temperature, sprayed with 0.2% *w*/*v* naphtoresorcinol in ethanol-H_2_SO_4_ (96:4, *v*/*v*), and heated at 105 °C for 10 min to visualize the saccharide bands [26].

### 2.6. Analysis of Monosaccharide Composition

The monosaccharide composition of the degraded polysaccharide was analyzed by using HPLC system (Agilent 1260, Santa Clara, CA, USA). The degraded polysaccharide was hydrolyzed at 120 °C with 1.3 M trifluoroacetic acid (TFA) for 3 h. After hydrolysis, the resulting sugar was labeled by 1-phenyl-3-methyl-5-pyrazolone (PMP). Then, the PMP derivatives were analyzed by HPLC-PAD with a Silgreen ODS C18 (250 × 4.6 mm, 5 μm) which was kept at 30 °C. The mobile phase was composed of 20 mM ammonium acetate-acetonitrile (83:17, *v*/*v*), and the flow rate was set as 1.0 mL/min [27].

### 2.7. Fourier-Transform Infrared (FT-IR) Spectroscopic Analysis

FT-IR spectra of the lyophilized degraded polysaccharides in KBr pellets were recorded at room temperature on a Spectrum One-B FTIR Spectrometer (Perkin Elmer, Waltham, MA, USA). The FTIR spectra were recorded in the frequency range of 500–4000 cm^−1^ and the resolutions were set as 4 cm^−1^.

### 2.8. Mass Spectrometry Analysis

The degraded polysaccharide was analyzed by HPLC-MS^n^ after derivation with PMP. HPLC-MS^n^ experiment was performed on a Finnigan LXQ ion trap mass spectrometer (Thermo Fisher, Pittsburgh, PA, USA) equipped with an electrospray ion source (ESI) and a photodiode array detector (PAD) controlled by XCalibur software (Thermo). A TSKgel-Amide-80 (20 × 150, 3 μm) column used to separate the PMP labeled saccharides. The analysis was conducted in positive mode, the spray voltage was set to +4.5 kV, and the scan range was set from *m*/*z* 100 to 2000. The capillary temperature was 275 °C and the mobile phase composed of 20 mM acetate–acetonitrile (83:17, *v*/*v*, pH 6) at a flow rate of 0.2 mL/min.

### 2.9. Anticoagulant Activity

All the coagulation experiments were performed according to the method reported by Mourᾶno et al. [28] on a coagulometer (infinite M200). APTT (Activated partial thromboplastin time), PT (Prothrombin time) and TT (Thrombin time) assays were performed using kits (Shanghai Solar Biotechnology, Shanghai, China). Briefly, 100 μL of pre-warmed APTT reagent was mixed with 80 μL of normal rabbit plasma and 20 μL of pre-warmed polysaccharide in 0.9% NaCl, after incubation at 37 °C for 3 min the mixture was added with 100 μL of 0.025 mol/L calcium chloride (37 °C) and then APTT was recorded as the time by the coagulometer. For prothrombin time (PT) assay, citrated normal rabbit plasma (80 μL) was mixed with 20 μL of polysaccharide in 0.9% NaCl solution, after the incubation at 37 °C for 1 min, 200 μL of pre-warmed PT assay reagent was added, and the clotting time was recorded. For TT assay, 160 μL of citrated normal rabbit plasma was mixed with 40 μL polysaccharide in 0.9% NaCl solution, followed by incubation at 37 °C for 1 min, and the clotting time was recorded after addition of 200 μL of pre-warmed TT.

### 2.10. Cell Culture Assay

Human colon adenocarcinoma cells (HT-29) were obtained from Type Culture Collection of Chinese Academy of Sciences (Shanghai, China). These cells were cultured in Dulbecco’s modified essential medium (DMEM) with high glucose, 1% amino acids, 10% fetal bovine serum (FBS), and 1% antibiotic.

### 2.11. Cytotoxicity Study in HT-29 Cells

The MTT (methyl thiazolyl tetrazolium) assay was used to investigate the cytotoxicity of degraded polysaccharide against HT-29 cells [8]. Briefly, 200 μL of culture medium containing 2 × 10^4^ cells were added to each well of 96-well plate. After 24 h culture, the medium was replaced with 100 μL culture medium containing test samples of different concentrations (200, 400, 600, 800, 1000 and 2000 μg/mL). Then the viability of the HT-29 cells was evaluated by MTT method after another 24 h. The absorbances of the wells were determined by using a microplate reader (Biotek, Winooski, VT, USA) at 570 nm. H_2_O_2_ (50 μg/mL) was used as positive control and the test samples (2000 μg/mL) that were added into the culture medium without HT-29 cells was used as blank control. Finally, the relative cell viability was calculated with the following equation.
Relative cell viability (%) = (A_sample_ − A_blank_)/ (A_control_ − A_blank_) × 100

### 2.12. Chromogenic Factor XII Activation Assay

Human activated coagulation factor XII (FXIIa) ELISA Kit was purchased from Wuhan Huamei Bioengineering Co., Ltd. (Wuhan, China) The chromogenic factor XII (FXII) activation assay of degraded fucoidans was conducted according to the kit instructions using standard human plasma. The absorbances of the wells were determined by using a microplate reader (Biotek, VT, USA) at 450 nm. Standard was used as a positive control and saline solution (0.9% NaCl) was used as a negative control.

### 2.13. Statistical Analysis

All the results were expressed as mean ± SD. Data were analyzed using SPSS 17.0 statistical software. Differences between the groups were considered statistically significant at *p* < 0.05.

## 3. Results and Discussions

### 3.1. Optimization of Photocatalytic Degradation Conditions

The photocatalytic degradation reaction was optimized by varying the illumination time and the addition amount of TiO_2_ and H_2_O_2_. As illustrated in the Figure 1A, the average molecular weight of fucoidan decreased gradually in the photocatalytic shown in Figure 1B, but it took 9 h to yield the product with the average molecular weight ≤ 10 kDa. A further photocatalytic reaction with 5% TiO_2_ as well as 0.95% H_2_O_2_ showed an even higher degradation efficiency, and the product had an average molecular weight of 3 kDa after 3 h of reaction (Figure 1C). Then the degradation results of the reaction with 0.24% and 0.48% of H_2_O_2_ were compared with that with 0.95% H_2_O_2_, and as illuminated in Figure 1D,E, 0.24% and 0.48% H_2_O_2_ could not reduce the average molecular weight of fucoidan to ≤ 10 kDa after 3 h of the reaction as 0.95% H_2_O_2_. In addition, fucoidan could not be decreased with 5% TiO_2_ and 0.95% H_2_O_2_ within 6 h in the absence of light (Appendix A).

TLC analysis demonstrated the low-molecular-weight products produced by the photocatalytic reaction (Figure 1G). After an hour reaction, oligosaccharides could be observed, and the low-molecular-weight products accumulated alone with the reaction time. Then monosaccharide and other small molecules were also produced as the by-products after 4 h reaction. Obviously, 0.5 h reaction could significantly reduce the molecular-weight of fucoidan, while a reaction time of 3 h is suitable to prepare oligosaccharides with fewer by-products. Thus, the samples degraded for 0.5 and 3 h, named as DF-0.5 and DF-3, respectively, were selected for subsequent chemical analysis.

TiO_2_, as the photocatalyst, could generate photoinduced electrons and positive holes under the irradiation of light, and these charged species can further generate free radicals [29]. H_2_O_2_ which could also produce radicals, has been applied widely in the degradation of polysaccharides [30]. However, it took a longer time (6–48 h) to degrade fucoidan even at a higher reaction temperature (≥50 °C) with more H_2_O_2_ (≥2%) in the presence of metal ions [31], and demonstrated a much lower degradation efficiency than the photocatalytic reaction with H_2_O_2_. H_2_O_2_ has been worked as an electron acceptor due to its high oxidizing efficiency as a consequence of its high oxidation potential and can promote an increase in the production of free radicals [32]. Other researchers have also found that the efficiency of the photocatalytic degradation was significantly improved by the combination of TiO_2_ and H_2_O_2_ [32,33]. The results show that the photocatalytic degradation method could improve the degradation efficiency of fucoidan. Thus, a combination of 5% TiO_2_ and 0.95% H_2_O_2_ was applied in the subsequent experiments in the present study.

### 3.2. Chemical Composition Analysis

The degradation products of fucoidan prepared under optimized conditions (1.0 g TiO_2_ and 6.4 mL H_2_O_2_ for 0.5 h and 3 h, named as DF-0.5 and DF-3, with yields of 87.38% and 43.34%, respectively) were subject to chemical composition analysis (Table 1). The average molecular weights of DF-0.5 and DF-3 were 90 kDa and 3 kDa, respectively. The chemical composition of DF-0.5 showed great similarity to that of fucoidan, while the contents of the total sugar and sulfated polysaccharides of DF-3 were obviously decreased compared with the original fucoidan.

To identify the functional groups of the degraded polysaccharide, FTIR spectra of Fucoidan, DF-0.5 and DF-3 were compared as shown in Figure 2. A major broad band at approximately 3440 cm^−1^ and an absorption band at approximately 1055 cm^−1^ were respectively attributed to the O-H and C-H stretching vibrations. The absorption band at around 1649 cm^−1^ was derived from the stretching vibration of the carboxylate anion of uronic acids, and the absorption band at around 1263 cm^−1^ was derived from the stretching vibration of S=O in the sulfate group. The FTIR spectra of Fucoidan, DF-0.5 and DF-3, showed great similarities and no obvious differences were observed. This indicates the photocatalytic degradation did not change the function groups of fucoidan.

Monosaccharide composition analysis was carried out by HPLC-PAD following acid hydrolysis and derivatization with PMP. As shown in Figure 2B, Fuc, DF-0.5 and DF-3 were all composed of galactose (Gal) and fucose (Fuc), but their ratio showed differences. Obviously, the mass ratio of galactose to fucose in DF-3 (0.7:1.0) was much lower than those in Fuc (1.3:1.0) and DF-0.5 (1.2:1.0). Thus, this result revealed that galactose in the polysaccharide structure was gradually reduced in the photocatalytic degradation.

Since many bioactivities of sulfated polysaccharides are related to their sulfate group, it is necessary to pay attention to the influence of photocatalytic degradation on the sulfate group content. The chemical composition analysis results indicate the sulfate group in the polysaccharide chain was not reduced during the photocatalytic degradation, and this phenomenon was also observed in the oxidation degradation with H_2_O_2_ [31]. The decrease of the sulfated polysaccharide content determined by DMB suggests the presence of sulfated small saccharides in DF-3 because the sulfated tri-, di- and mono-saccharide was undetectable in the DMB assay. In addition, the decrease of galactose in DF-3 indicates that galactose in the chain was more ready to break down into monosaccharide or other small molecules than fucose in the photocatalytic degradation reaction.

### 3.3. Identification of Oligosaccharides Produced by Photocatalytic Degradation

The oligosaccharides in DF-3 were analyzed by HPLC-ESI-MS^n^ after PMP derivatization, and tetra-, tri- and di-saccharides were detected and elucidated by their MS data. The ratios of the peak areas of the identified oligosaccharides in extracted ion chromatograms were shown in Appendix A and the pseudo-molecular ions of PMP-labeled oligosaccharides from the DF-3 were illustrated in Appendix A.

As illustrated in the Figure 3A, the pseudo-molecular ion of the Peak 1 at *m*/*z* 1143 [M+H]^+^ gave product ions at *m*/*z* 1063 [M-SO_3_+H]^+^, *m*/*z* 983 [M-2SO_3_+H]^+^, *m*/*z* 903 [M-3SO_3_+H]^+^, *m*/*z* 837 [M-3SO_3_-Fuc+H]^+^ and *m*/*z* 757 [M-4SO_3_-Fuc+H]^+^ in MS^2^. Thus, Peak **1** was determined as a tetrasaccharide composed of 3 sulfate groups, 3 Fuc residues and a 134 Da residue which was inferred as 2,3,4-trihydroxypentanal, a photocatalytic degradation product of Fuc. The possible photocatalytic degradation process of Fuc to yield 2,3,4-trihydroxypentanal (dFuc) is shown in Figure 4. Due to the attack of the radicals in the reaction solution, the aldehyde at C-1 of Fuc was oxidated to carboxylic acid, and then the decarboxylation reaction yielded a pentane-1,2,3,4-teraol. Subsequently, the oxidation of the hydroxyl group to aldehyde group produced the final product, 2,3,4-trihydroxypentanal.

As shown in Figure 3B–E, some kinds of trisaccharides were observed in DF-3. The pseudo-molecular ion of Peak **2** at *m*/*z* 997 [M+H]^+^ gave product ions at *m*/*z* 917 [M-SO_3_+H]^+^, 837 [M-2SO_3_+H]^+^, 771 [M-2SO_3_-Fuc+H]^+^, 691 [M-3SO_3_-Fuc+H]^+^, 611 [M-4SO_3_-Fuc+H]^+^ and 495 [M-4SO_3_-2Fuc+H]^+^ in MS^2^. So Peak **2** was identified as the PMP derivative of tetra-sulfated Fuc→Fuc→dFuc. Peaks **3–8** in Figure 3C all had pseudo-molecular ions at *m*/*z* 917, and offered similar MS^2^ spectra (Appendix A). They all showed product ions at *m*/*z* 837 [M-SO_3_+H]^+^ in MS^2^ by losing a SO_3_ residue, and they also gave fragment ions 771 [M-2SO_3_-Fuc+H]^+^, 691 [M-3SO_3_-Fuc+H]^+^, 611 [M-4SO_3_-Fuc+H]^+^ and 495 [M-4SO_3_-2Fuc+H]^+^ as Peak 5. Therefore, Peaks **3–8** were identified as PMP labeled tri-sulfated Fuc→Fuc→dFuc. The pseudo-molecular ions of Peak **9–12** all showed at *m*/*z* 947 [M+H]^+^ which gave product ions at *m*/*z* 867 [M-SO_3_+H]^+^, 787 [M-2SO_3_+H]^+^, 641 [M-2SO_3_-Fuc+H]+ and 495 [M-2SO_3_-2Fuc+H]^+^ in MS2. Peaks **13** and **14** demonstrated their pseudo-molecular ion at *m*/*z* 867 [M+H]^+^, and their structure could be confirmed by fragment ions at *m*/*z* 787 [M-SO_3_+H]+, 641 [M-SO_3_-Fuc+H]^+^ and 495 [M-SO_3_-2Fuc+H]^+^. Thus, Peaks **9–12** in Figure 3D and Peaks **13** and **14** in Figure 3E were established as PMP derivatives of di-and mono-sulfated Fuc→Fuc→Fuc, respectively.

As illustrated in the Figure 3F–L, some disaccharides could also be observed in DF-3. Peak **15** in Figure 3F shows a pseudo-molecular ion at *m*/*z* 815 [M+H]^+^ which affords product ions at *m*/*z* 735 [M-SO_3_+H]^+^, 655 [M-2SO_3_+H]^+^, 589 [M-SO_3_-Fuc+H]^+^ and 509 [M-2SO_3_-Fuc+H]^+^ in MS^2^. The disaccharide was composed of 2 sulfate groups, a Fuc residue and a 160 Da residue which was inferred as dehydrogenated Gal (dGal) produced from the photocatalytic degradation of Gal. Besides, Peaks **16** and **17** in Figure 3G with the same pseudo-molecular ions at *m*/*z* 895 [M+H]^+^ offered fragment ions shown at *m*/*z* 815 [M-SO_3_+ H]^+^ and *m*/*z* 735 [M-2SO_3_+H]^+^, indicating an additional SO_3_ residue compared to Peak **15**. Peaks **16** and **17** were identified as PMP derivatives of trisulfated Fuc→dGal.

Peaks **18–20** in Figure 3H show the same pseudo-molecular ion at *m*/*z* 771 [M+H]^+^ which affords a product ion at *m*/*z* 691 [M-SO_3_+H]^+^, 611 [M-2SO_3_+H]^+^ and 465 [M-2SO_3_-Fuc+H]+, indicating its structure as PMP labeled di-sulfated Fuc→dFuc. Peaks **21** and **22** in Figure 3I with uniform pseudo-molecular ions at *m*/*z* 721 [M+H]^+^, further yielded fragment ions at *m*/*z* 641 [M-SO_3_+H]^+^ and *m*/*z* 495 [M-SO_3_-Fuc+H]^+^. Thus, Peaks **21** and **22** were both identified as PMP labeled mono-sulfated Fuc→Fuc, but they differed in the sulfated substitution. Peak **23** in Figure 3J with a pseudo-molecular ion at *m*/*z* 691 [M+H]^+^ was characterized as a PMP labeled mono-sulfated Fuc→dFuc by its fragment ions at *m*/*z* 611 [M-SO_3_+H]^+^ and *m*/*z* 465 [M-SO_3_-Fuc+H]^+^. The pseudo-molecular ion of Peak **24** at *m*/*z* 673 [M+H]^+^ provides product ions at *m*/*z* 511 [M-Gal+H]^+^ which were characteristic for a disaccharide of Gal→Gal. Peak **25** in Figure 3M shows a pseudo-molecular ion at *m*/*z* 627 [M+H]^+^ which affords a product ion at *m*/*z* 465 [M-Gal+H]^+^. Thus, the saccharide portion of Figure 3L was identified as PMP labeled Gal→dGal. In the present study, a series of oligosaccharides, including di-, tri- or tetra-saccharides, were observed in the photocatalytic degradation products of fucoidan, and interestingly, sulfate groups remained in oligosaccharide fragments. Some methods have already been applied to degrade sulfated polysaccharides, such as acid hydrolysis and hydrogen peroxide degradation. However, acid hydrolysis could remove the sulfate groups which is important for the bioactivity and function of the sulfated polysaccharides [34]. Hydrogen peroxide degradation [31] failed to produce oligosaccharides. The findings in the present study suggest that photocatalytic degradation could retain the sulfate group of fucoidan and have more advantages in the production of sulfated oligosaccharides from polysaccharides.

### 3.4. Cytotoxicity of the Degraded Fucoidans

Considering the emergence of novel fragments in the photocatalytic degradation, it is necessary to evaluate the toxicity of the degradation products of fucoidan for the safety in their further application. Numerous researches have revealed that fucoidan could reduce the HT-29 cell proliferation or induce apoptosis of HT-29 cells [35,36]. Then, in the present study, the influences of fucoidan, DF-0.5 and DF-3 on the proliferation of HT-29 human colon adenocarcinoma cells were determined by the MTT method. As shown in Figure 5, all the three samples showed no significant cytotoxicity at concentrations ≤2000 μg/mL. Numerous researches have revealed that many polysaccharides could reduce the cell proliferation at a concentration of around 1000 μg/mL [37,38]. Fucoidan, it has also been reported, could reduce the number of viable Lewis Lung Carcinoma and melanoma B16 cells at a concentration of around 1000 μg/mL [39]. The findings in the present study suggested that photocatalytic degradation could not enhance the cytotoxicity of fucoidan, indicating the safety of the photocatalytic degradation products of fucoidan.

### 3.5. Anticoagulant Activity of the Degraded Fucoidans

Fucoidan has been found to have significant anticoagulant activity which indicates its potential application as an anticoagulant as well as the risk to cause hemorrhage in its other applications. In the present study, the anticoagulant activities of fucoidan, DF-0.5 and DF-3 were evaluated by the APTT, PT and TT assays using heparin as a reference. As shown in Figure 6, DF-0.5 could prolong APTT, PT and TT in a concentration-dependent manner, although its anticoagulant activity was similar to fucoidan, but weaker than that of heparin. However, DF-3 only prolonged the clotting time of APTT obviously but its capability was weaker than fucoidan and DF-0.5. Moreover, DF-3 showed no significant effect on the clotting time of PT and TT. Given the fact that fucoidan, DF-0.5 and DF-3 have similar sulfate contents but vary in the molecular weight and the molar ratio of fucose to galactose, this finding is consistent with the previous report that molecular weight and the molar ratio of fucose to galactose plays a vital role in the anticoagulant activity of fucoidan [40]. As we know, TT indicates the inhibition of the common pathways of coagulation, while APTT and PT indicate the inhibition of the intrinsic pathway of coagulation and extrinsic pathway, respectively [41]. Interestingly, DF-3 specifically prolonged the clotting time of only APTT but not PT or TT, suggesting its selective participation in the intrinsic pathway of the blood coagulation cascade.

The effects of fucoidan, DF-0.5 and DF-3 on FXII which triggers the intrinsic pathway of blood coagulation were further investigated in vitro. As shown in Figure 7, all samples exerted inhibition effects on the intrinsic pathway FXII in a dose-dependent manner. Fucoidan and DF-0.5 exerted stronger inhibition effects on FXII than DF-3. These results were consistent with the results of the APTT, PT and TT assays. Notably, DF-3 could specifically prolong the clotting time of APTT, but not PT or TT, and it also exerted inhibition effects on the FXII. This indicates DF-3 targets the intrinsic pathway of coagulation although its effect is not as strong as fucoidan or DF-0.5. It has been reported that FXII inhibition could reduce thrombosis without causing abnormal bleeding [42]. Thus, the photocatalytic degradation of fucoidan is promising to provide safer anticoagulant agents targeting the intrinsic coagulation pathway without increasing the bleeding risk.

## 4. Conclusions

The photocatalytic degradation reaction condition was optimized to obtain low-molecular-weight fucoidans by varying the illumination time and the addition amounts of TiO_2_ and H_2_O_2_. The composition and structure of the degraded fucoidan were characterized, and the results indicated that the photocatalytic degradation did not remove the sulfate groups of fucoidan, and sulfated tetra-, tri-, and di-saccharides were identified in DF-3. All the photocatalytic degradation products showed no significant cytotoxicity at concentrations ≤2000 μg/mL. Moreover, fucoidan and DF-0.5 could prolong APTT, PT and TT, but DF-3 could only prolong the clotting time of APTT. All samples exerted inhibition effects on the intrinsic pathway FXII in a dose-dependent manner. Thus, the present study demonstrated the photocatalytic degradation as an environmentally friendly, low-cost and effective method to prepare low-molecular-weight fucoidans, but not stripping their sulfate groups.

## Figures and Tables

**Figure 1 foods-11-00822-f001:**
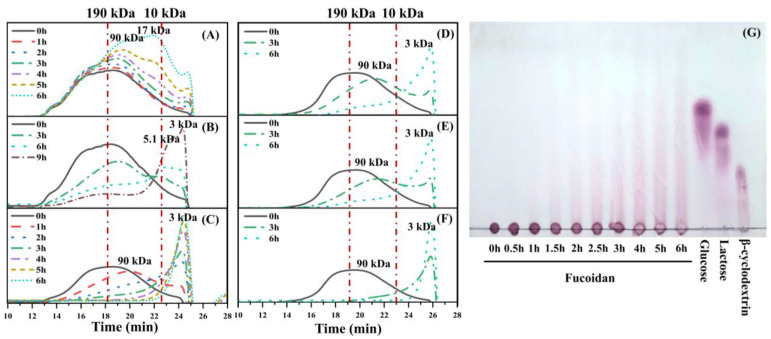
Molecular weight distribution of fucoidan degraded with 1% TiO_2_ (**A**), 5% TiO_2_ (**B**), 5% TiO_2_ and 0.95% H_2_O_2_ (**C**), 5% TiO_2_ and 0.24% H_2_O_2_ (**D**), 5% TiO_2_ and 0.48% H_2_O_2_ (**E**), 5% TiO_2_ and 0.95% H_2_O_2_ (**F**), and TLC of degraded fucoidan (**G**).

**Figure 2 foods-11-00822-f002:**
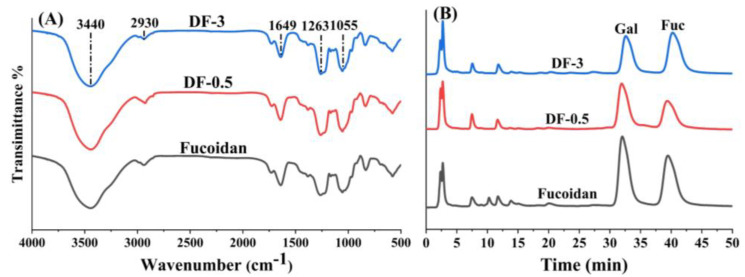
FT-IR spectra of freeze-dried samples (**A**) and HPLC chromatograms of the component monosaccharides (**B**) of fucoidan and its 2 degradation products (DF-0.5 and DF-3).

**Figure 3 foods-11-00822-f003:**
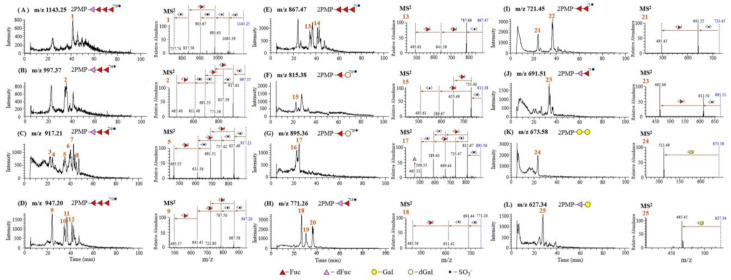
Extracted-ion chromatograms and MS^2^ of PMP-labeled oligosaccharides in the DF-3.

**Figure 4 foods-11-00822-f004:**
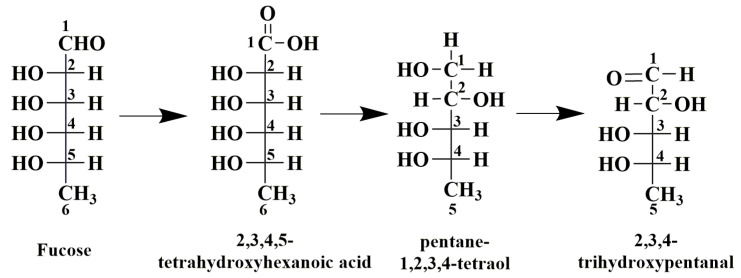
The chemical structure change of fucose during photocatalytic degradation reaction.

**Figure 5 foods-11-00822-f005:**
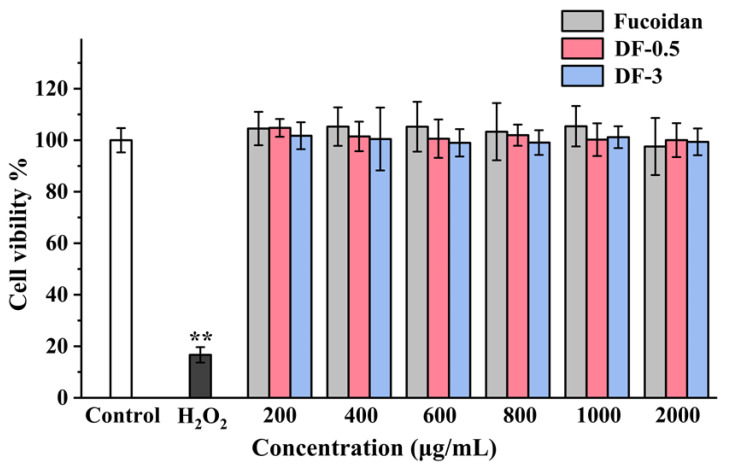
Effects of the degraded fucoidans on the HT-29 cell proliferation. Results were presented as mean ± SD (n = 6). ** *p* < 0.01 indicate statistically significant differences from the control group.

**Figure 6 foods-11-00822-f006:**
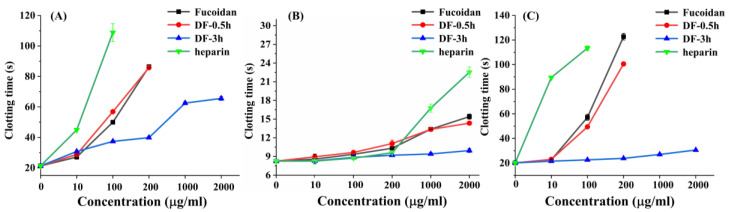
Comparison of the activated partial thromboplastin time (**A**), prothrombin time (**B**) and thrombin time (**C**) of the degradation products (DF-0.5 and DF-3) with those of fucoidan and heparin (n = 4).

**Figure 7 foods-11-00822-f007:**
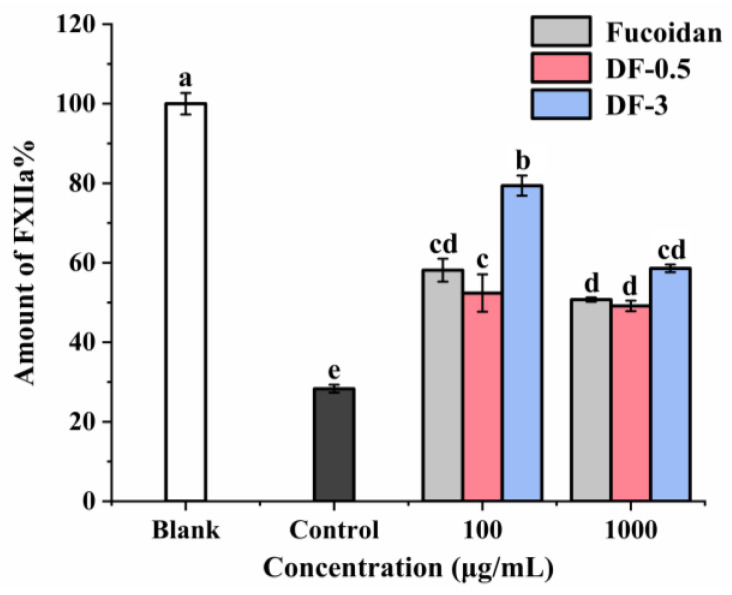
Effects of degraded fucoidans on the coagulation factor XII. Results were presented as mean ± SD (n = 2). The small letters (a–e) indicate the significant difference (*p* < 0.05).

**Table 1 foods-11-00822-t001:** Chemical compositions of fucoidan and its 2 degradation products (DF-0.5 and DF-3).

Sample	Average Mw/kDa	Total Sugar %	Sulfate Group %	Sulfated Polysaccharides %
Fucoidan	190	47.52 ± 1.93	26.22 ± 2.04	97.10 ± 0.01
DF-0.5	90	48.46 ± 1.00	23.08 ± 0.92	95.85 ± 0.04
DF-3	3	36.46 ± 2.49 **	22.69 ± 0.72	57.70 ± 0.01 ***

Data are expressed as mean ± SD (n = 3). ** *p* < 0.01 and *** *p* < 0.001 indicate statistically significant differences from the control group.

## Data Availability

Not applicable.

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
