# Peer review of "Preparation of Low-Molecular-Weight Fucoidan with Anticoagulant Activity by Photocatalytic Degradation Method"

_foods, 2022, doi:10.3390/foods11060822_

Round 1

Reviewer 1 Report

Overview  

        The authors performed a comprehensive investigation of the preparation of low-molecular-weight fucoidan with anticoagulant activity by the photocatalytic degradation method. In this study, the author clearly described the TiO2-catalyzed photocatalytic reaction for fucoidan degradation, and the degradation products were characterized and compared with the un-degraded fucoidan by a series of analyses techniques to reveal the compositional, structural, cytotoxicity, and anticoagulant activity changes of polysaccharides caused by photocatalytic degradation. The results suggest that the photocatalytic degradation demonstrated the potential to prepare sulfated low-molecular-weight fucoidan with anticoagulant activity. The authors put forth a lot of effort to complete the research work, with enough experiments and well-analyzed data. There are a few suggestions to increase its quality.

General comments 

Presentation: Satisfied.

Introduction: It is looking good

Materials and Methods: Presentation is appropriate however it should elaborate in details

Results and Discussion: Expressed with suitable explanations and written well however this part must be strengthened. 

Conclusion: It is looking good

Reference: The author should cross-check the reference

Language: Good

 Comments

  1. In the abstract section, the abbreviation should be explained. i.e. HPGPC and TLC.
  2. In the material and method section, the author should modify the 2.1 chemicals into materials.
  3. In the 2.2 section, the author should mention the lyophilization details.
  4. In the 2.2. section, the author should follow any specific pH for photocatalytic degradation of fucoidan?
  5. In the 2.3 chemical analysis, the author failed to mention the standard for sulfate analysis.
  6. In the 2.4 section, the author should include dextran purchase details.
  7. In the 2.5 thin layer chromatography (TLC) section, the author needs to include different time details.
  8. In 2.7, the author should mention the spectrum range. (i.e., 500 – 4000 cm-1).
  9. In the cell culture assay, the author should explain the HT-29 cells into human colon adenocarcinoma cells (HT-29) or HT-29 (human colon adenocarcinoma cells) and include the different concentrations of the sample (200-2000 µg/ml).
  10. Figure 1, the author must include different hours’ details in B-F images.
  11. Figure 6, the specifications (Fucoidan, DF-0.5h, DG-3h, and heparin) were missing in the A and B images.
  12. The author should check the scientific name in the entire manuscript and it should be italic.

Reviewer 2 Report

The authors proposed an interesting approach to obtaining low-molecular-weight fucoidan with anticoagulant activity by photocatalytic degradation method. Given the high interest of researchers in this topic, I am sure that the results of these studies will be widely demanded. For me, the choice of Foods to publish the results of this study is not obvious, but it does not contradict the Aims & Scope of this journal.

Here are my comments/questions to improve the present version of the manuscript:

- HT-29 cell line has been selected as model cell line. Why this one? A brief statement justifying this selection would be welcome.

- A scheme illustrating the first paragraph of the RESULTS/DISCUSSION section would be helpful.

- The authors suggest that the synthesized compounds can be introduced into medical practice as anticoagulants, meanwhile, the pharmaceutical industry pays special attention to standardization and control of the constancy of the composition, how reproducible the experiments are and how much the properties of the final product differ depending on the source and time of collection of the raw material (fucoidan)?

- Described in the column "Supplementary Materials" are not available to the reviewer.
